

# 1 2 Foraminiferal holobiont thermal tolerance under climate change-Roommates problems or successful collaboration?

Doron Pinko, Sigal Abramovich, and Danna Titelboim
The Department of Geological and Environmental Sciences, Ben-Gurion University of the Negev, Beer Sheva, 8410501,
Israel.
*Correspondence to*: Danna Titelboim (dannati@post.bgu.ac.il)

## 7 Abstract

Understanding the response of marine organisms to expected future warming is essential. Large Benthic Foraminifera (LBF)
are symbiont bearing protists considered to be major carbonate producers and ecosystems engineers. We examined the
thermal tolerance of two main types of LBF holobionts characterized by different algal symbionts and shell types (resulted
from alternative biomineralization mechanisms): The hyaline diatom bearing, *Amphistegina lobifera*, and the porcelaneous
dinoflagellate bearing, *Sorites orbiculus*. To assess the relative contribution of host and symbiont algae to the holobiont
thermal tolerance we separately evaluated their response by measuring calcification rates and photosynthetic activity under
present-day and future warming scenarios. Our results show that both holobionts exhibit thermal resilience up to 32°C and
sensitivity to 35°C. This sensitivity differs in the magnitude of their response: calcification of *A. lobifera* was completely
inhibited while it was only reduced in *S. orbiculus*. Thus, future warming will significantly shift the relative contribution of
the two species as carbonate producers. Moreover, *A. lobifera* exhibited a synchronized response of the host and symbionts.
In contrast, in *S. orbiculus* the symbionts responded prior to the host, possibly limiting its resilience. Our results also
demonstrate the role of pre-exposure and acclimation processes of host, symbionts or both in mitigating future warming. It
highlights the possibility that while pre-exposure to moderate temperatures benefits the holobiont, in cases of extreme
temperature it might reduce its thermal tolerance.

## 22 1 Introduction

Since the beginning of the industrial revolution anthropogenic activity has been leading to rapid ocean warming. This
negatively affects marine ecosystems and specifically symbiont bearing calcifiers (Kawahata et al., 2019). The observed rate
of global Sea Surface Temperature (SST) rise stands on 0.11°C per decade and future scenario predicts a similar rate until
the end of the century (IPCC, 2014). Warming in the Eastern Mediterranean is expected to rise almost four times more
rapidly than global forecast (Macias et al., 2013). Thus, the Eastern Mediterranean is expected to be one of the most affected
regions by global warming. Therefore, this area can be presented in biogeographic models as a "miniature ocean" providing
indications on global patterns in marine ecosystems in a warmer world (Lejeusne et al., 2010).





Symbiont bearing Large Benthic foraminifera (LBF) are single-celled ecosystems engineers. Their carbonate production is
estimated as at least 5% of the annual production in reef and carbonate shelf environments (Langer, 2008; Langer et al.,
1997). Temperature is a major factor in the distribution of LBF that exhibit distinct thresholds for reproduction, survival,
bleaching, and calcification (Evans et al., 2015; Hallock et al., 2006a; Langer et al., 2012; Langer and Hottinger, 2000;
Schmidt et al., 2011; Titelboim et al., 2019; Weinmann et al., 2013). The symbiont composition of LBF appears to be
strongly controlled by temperatures (Momigliano and Uthicke, 2013; Prazeres, 2018; Prazeres et al., 2017; Prazeres and
Renema, 2019; Schmidt et al., 2018) which explains the observations that species-specific thermal tolerance is associated
with more diverse algal symbionts (Stuhr et al., 2018).
Many LBF species are Lessepsian invaders, which often comprise over 90% of the foraminiferal population in the Eastern
Mediterranean (Hyams-Kaphzan et al., 2014; Titelboim et al., 2016). Their invasion and successful establishment are
facilitated by rising temperatures, as in the case of other Lessepsian organisms (Por, 1978, 2010; Zenetos et al., 2010, 2012).
However, some of these species currently live very close to their thermal thresholds and consequently, their presence will be
impeded in the relatively near future (Titelboim et al., 2016). The thermal sensitivity of some LBF species has already been
investigated (Schmidt et al., 2011, 2016b; Stuhr et al., 2018; Titelboim et al., 2019). Yet, the relative contribution (positive
or negative) of the host and symbiont algae to cope with rising temperatures has not been fully constrained.
In this study, we present the thermal sensitivity of two very dominant and prominent calcifiers LBF species with different
holobiont systems. Specifically, our study separately assesses the thermal sensitivity of the foraminiferal host and algal
symbiont by tracking their calcification rate and photosynthetic activity as an indication of their well-being under different
warming scenarios (Fig. 1).

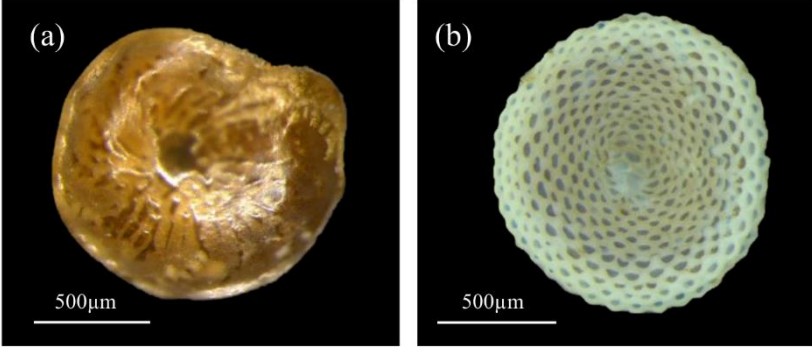


**Figure 1: The holobionts examined in this study. a)** *Amphistegina lobifera* **and b)** *Sorties orbiculus***. Note the green-brownish color**
**of the symbiont algae.**



## 2 Materials and methods

### 2.1 Specimens collection and handling

In this study, we targeted two LBF species that represent different types of holobiont systems, which differ in their shell construction mechanism and algal symbionts: *Amphistegina lobifera* (diatom bearing hyaline, Schmidt et al., 2016b) and *S. orbiculus* (dinoflagellate bearing porcelaneous, Merkado et al., 2013). Both species have cosmopolitan distributions, are very common in warm shallow marine environments (Langer and Hottinger, 2000) and display different thermal tolerances (Titelboim et al., 2016). Specimens were picked from macro-algal samples that were scraped from beach rocks at Shikmona, northern Mediterranean coast of Israel during February and May 2019. To reduce variance in growth derived from ontogenetic variability, the specimens were picked between specific size fractions (see details in Supplement 1 Table S1). Live specimens (indicated by their symbiont color and motility) were cleaned by brushing, divided into groups with an equal number of individuals (details in Supplement 1 Table S1), and transferred into 60 ml airtight Erlenmeyer flasks, from here on referred to as 'samples'.

During the experiments, the samples were placed in temperature-controlled water baths, which maintained constant temperatures of ± 0.5°C, temperatures were monitored regularly. During the cultivating period, the samples were kept under a daily cycle of 12 hours light / 12 hours dark using fluorescent light of ~30 μmol photons m$^{-2}$ s$^{-1}$. These are lower than the photosynthetic optimum for *A. lobifera* (Ziegler and Uthicke, 2011), however, they were used to produce comparable data to that of related published papers (Schmidt et al., 2016b, 2016a, 2018; Titelboim et al., 2019).

### 2.2 Laboratory manipulative experiments

The temperature manipulative experiments were conducted separately on specimens collected during February and May 2019 (i.e. winter and spring populations). The February 2019 experiment examined only the calcification rate of *S. orbiculus*. Next, we conducted another experiment on *S. orbiculus* in May 2019 to examine possible variation in thermal tolerance between winter and spring populations (following Schmidt et al., 2016a). The temperature manipulative experiments on *A. lobifera* were conducted only on the spring population that was sampled in May 2019. In this experiment, the well-being of both holobionts was examined by separately determining the responses of the foraminiferal hosts (calcification rate) and their symbiont algae (photosynthetic activity) to elevated temperatures.

All samples were acclimated under constant conditions for at least ten days. Then, the calcification rate (February and May populations) and photosynthetic activity (May populations) were measured to establish the performance baselines of the different species and the natural variability between samples, under equal conditions. Samples that did not exhibit apparent photosynthetic activity (i.e. oxygen production) were excluded from the rest of the study. At the end of the acclimation period, seawater was replaced in all samples and the temperature of each bath was slowly adjusted (1°C/hour). The examined treatments (25°C, 30°C, 32°C, 35°C) represent current and future temperatures expected until the end of the century (Macias et al., 2013). Each temperature treatment included three to five replicates (Supplement 1 Table S1).

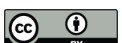



After acclimation, the samples were exposed to the designated temperatures for a total duration of three (May experiment) or
four (February experiment) weeks. After each week the water was replaced with fresh natural seawater.  Calcification rates
($\mu$mol CaCO$_3$ week$^{-1}$ specimen$^{-1}$) were calculated using the Alkalinity Anomaly Method (Smith and Key, 1975). In this
method, the calcification rate is determined from the change in total alkalinity of the seawater caused by the precipitation of
CaCO$_3$. These are determined by comparison to a control sample containing no foraminifera. Accuracy was assessed by
analyses of the Scripps Institute of Oceanography reference seawater (Batch 154, February and Batch 180, May) and an
internal standard.
Photosynthetic activity ($\Delta$O$_2$ $\mu$g/L specimen$^{-1}$) was measured as oxygen production compared with a control sample
containing no foraminifera. Dissolved oxygen was measured using RDO optical dissolved oxygen sensor. Accuracy was
assessed by calibration of the sensor against Winkler titration.
**2.3 Statistical analysis**
To examine whether differences in calcification rates and photosynthetic activity are significant between temperature
treatments and between weeks, statistical analyses were performed using STATISTICA10 software. For each set of data, we
tested assumptions of normality of the residuals and homogeneity of variances and a statistical test was chosen accordingly.
If both assumptions were valid ANOVA was performed, in cases were normality was valid and homogeneity was violated
Welch's ANOVA test was applied. In cases were normality was violated the non-parametric test Kruskal-Wallis was
applied. Each was followed by the proper post-hoc test.
**3 Results**
Our experimental design takes into consideration biological variability in calcification rates and photosynthetic activity
between different species and populations. This notion is based on previous observations that different species even from the
same genus, and different populations of the same species display different calcification rates under the same conditions (i.e.
baseline, Titelboim et al., 2019). Indeed, the activity baseline of both the hosts and the symbionts are significantly different
between *A. lobifera* and *S. orbiculus* and between winter and spring populations (Fig. 2, Supplement 1 Tables S2 and S3).
Hence, the thermal tolerance of the two holobionts was separately evaluated for each experiment.





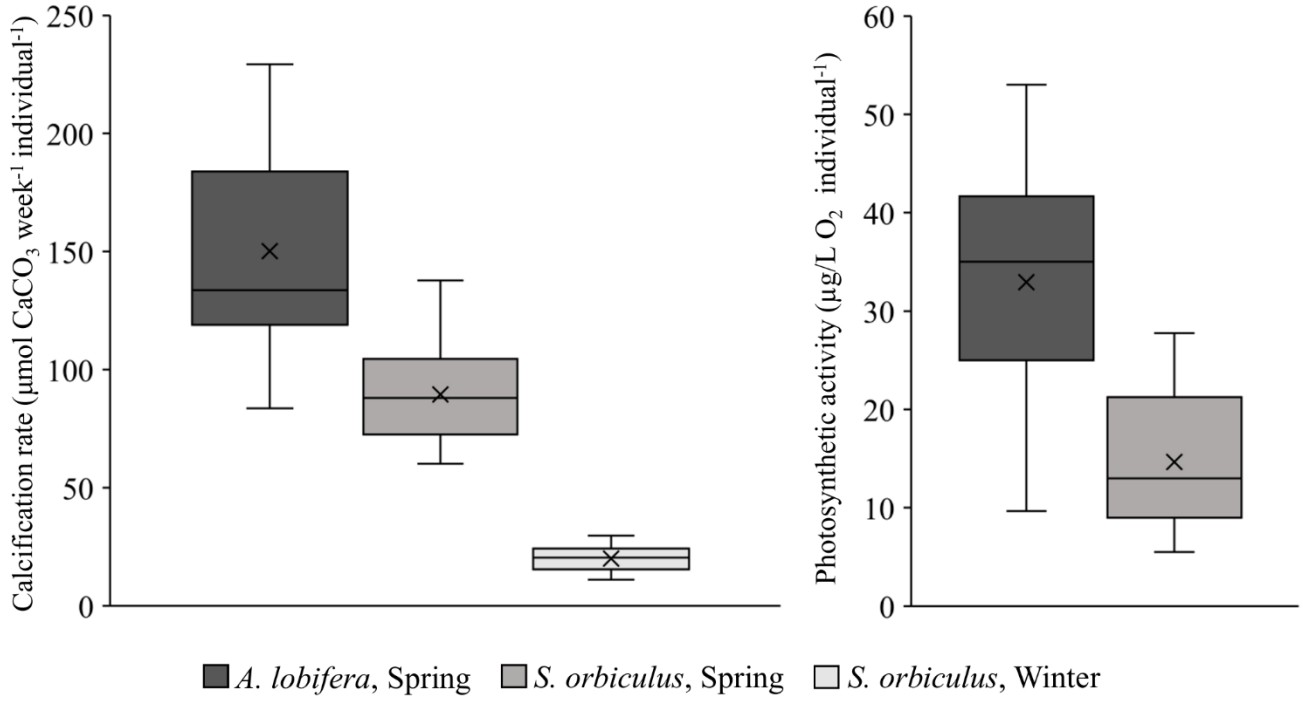

**Figure 2: Activity baseline of the hosts and symbionts indicated by calcification rates (left) and photosynthesis activity (right), respectively. Note, the distinct differences in baseline values of both calcification rates and photosynthetic activity between the two holobionts and between winter and spring populations of *S. orbiculus*.**

### 3.1 *Sorites orbiculus* (porcelaneous- dinoflagellate holobiont system)

Calcification rates of the spring population are much higher than those of the winter population indicated both in the baseline measurements (Fig. 2) and in the experiments (Fig. 3). Comparison between their calcification responses under the different temperature treatments reveals overall similar trends of highest values at 25°C, 30°C, and 32°C and a decrease at 35°C. The negative response at $35^{0}$C is substantially different between the populations: in the winter population, calcification decreases already after one week and is inhibited after three weeks (Fig. 3). Whereas in the spring population the calcification rate is reduced only after two weeks, and then remains low, but is not inhibited (Fig. 3, for statistical analyses, see Supplement 1 Tables S4 and S5).

Symbionts photosynthetic activity of the spring population indicates different thermal sensitivity patterns than that of the host. Throughout the experiment, positive values were observed under 25°C, 30°C, and 32°C. At 35°C, net photosynthesis was negative and gradually decreased during the experiment (Fig. 3, for statistical analyses, see Supplement 1 Table S6). Unaccountably, one sample exhibited an abnormal value (i.e. extreme in Fig. 3) with respect to previous weeks as well as to other replicates and thus was not included in the average and error calculations as it is clearly damaged from sample handling.





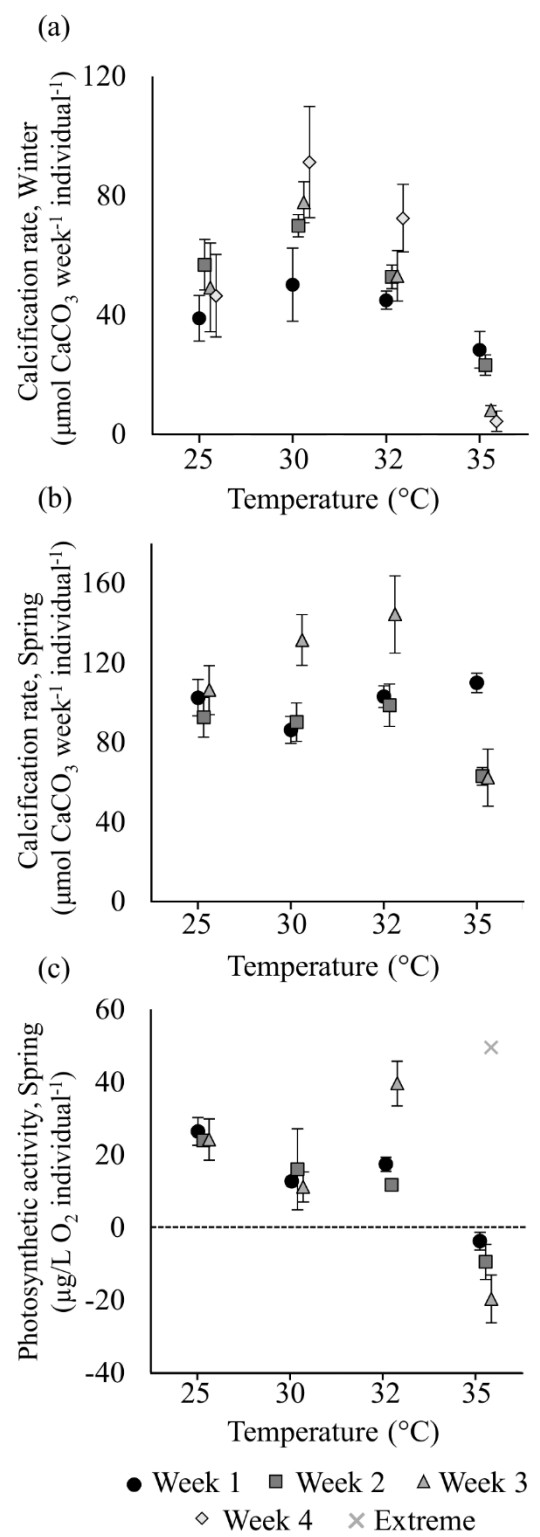





**Figure 3: Calcification rates (a, b) and photosynthetic activity (c) of *S. orbiculus* winter population (a) and spring population (b, c).**
**Note, the significant negative response of host and symbionts at 35°C (a-c) with the exception of the spring population calcification**
**rate at week 1 (b). Abnormal measurement is marked as extreme and is not calculated as part of the average and error.**

## 3.2 *Amphistegina lobifera* (hyaline diatom holobiont system)

Both calcification and photosynthesis responses remain synchronized throughout the experiment. After the first and second
weeks, calcification rates and photosynthetic activity exhibited the highest values under 25°C, 30°C, and 32°C. At 35°C
calcification and photosynthesis were both inhibited and net photosynthesis was negative (Fig. 4, Supplement 1 Tables S7
and 8). Between the second and third weeks, many specimens exhibit massive bleaching. The bleaching occurred in different
treatments between 25°C-32°C and thus clearly not related to the different temperature treatments. For this reason,
measurements of the third week are excluded from the results.

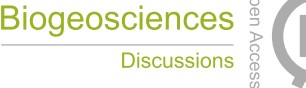



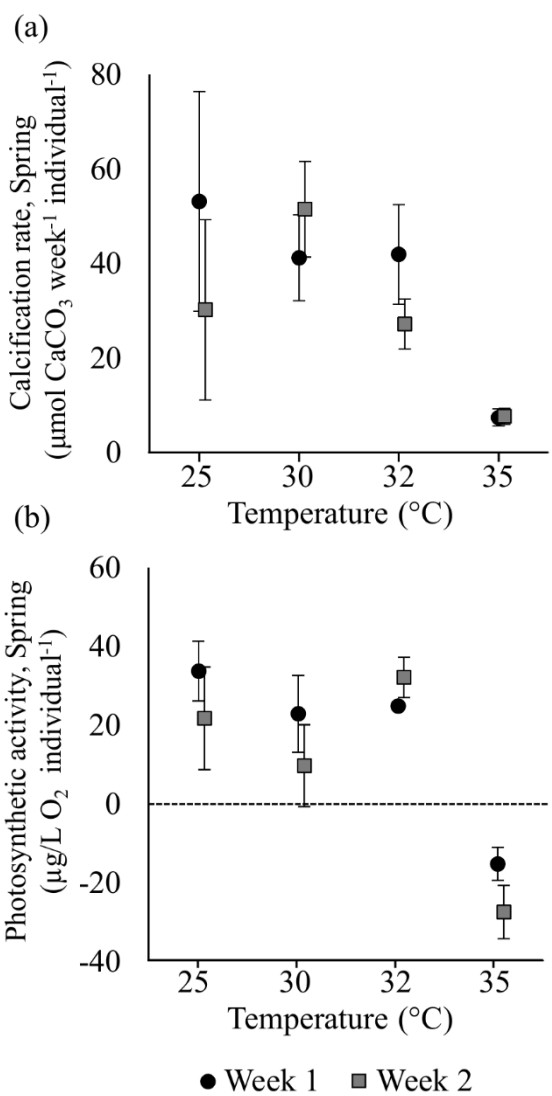

Figure 4: Calcification rates (a) and photosynthetic activity (b) of *A. lobifera* spring population. Note the synchronized significant negative response of both host and symbiont at 35°C.

## 4 Discussion

*Amphistegina lobifera* and *S. orbiculus* are both considered as prominent calcifiers based on their massive occurrences and widespread distribution (Langer and Hottinger, 2000). Our study reveals clear differences in their thermal tolerance as expressed by both calcification rates and their algal performance. Specifically, our results predict that with rising temperatures the relative contribution of *S. orbiculus* will increase since its calcification is not inhibited even at extreme





temperatures, contrary to *A. lobifera*. This highlights the need for species-specific considerations for more accurate
predictions on the fate of LBF and their future contribution to global carbonate production.
It was previously shown that corals ability to cope with elevated temperatures is strongly dependent on their partnering with
functionally diverse symbionts (Baker et al., 2004; Howells et al., 2012; Jones et al., 2008; Rowan, 2004), although their
symbiosis is limited to dinoflagellate from the *Symbiodinium "*Clades*"*(LaJeunesse et al., 2018; Silverstein et al., 2015).
LBF are known to host different kinds of symbionts (Pochon et al., 2007), which include dinoflagellates, diatoms, unicellular
chlorophytes, unicellular rhodophytes and/or cyanobacteria (reviewed in Lee, 2006). Whereas the general types of the
symbiont (algal genus) seem to be phylogenetically fixed, there appears to be considerable flexibility in symbiont infestation,
even within one individual (Lee, 2006). This versatile symbiont partnership may control holobionts thermal tolerance and
provide one of the key factors in their response to future warming. For example, a mechanism to cope with thermal stress
was observed in *Pararotalia calcariformata*, an extremely heat tolerant symbiont bearing foraminifera, by 'shuffling' of
symbiont communities (Schmidt et al., 2018). This might also explain an observation that species-specific differences in the
thermal tolerance of *Amphistegina* species are related to different symbiont assemblages. Specifically, a larger diversity of
algal symbionts was associated with the more tolerant species (Stuhr et al., 2018).
Our study separately describe the thermal sensitivity of the host and the symbionts in two types of holobiont systems: *A.*
*lobifera* hosting diatoms mostly from the order Fragilariales (Barnes, 2016; Prazeres et al., 2017; Schmidt et al., 2016b,
2018) and *S. orbiculus* hosting dinoflagellate, Symbiodinium (Merkado et al., 2013; Pawlowski et al., 2001; Pochon et al.,
2007). Both holobionts show thermal resilience up to 32°C and a negative response at 35°C (Figs. 3-4). Yet, they
substantially differ in respect to the magnitude and timing of their responses: *A. lobifera* and its diatom symbionts share
similar thermal sensitivity with inhibition of calcification and negative net photosynthesis at 35°C. Whereas in *S. orbiculus*
calcification is reduced but not inhibited indicating it is more resilient to extreme SST than *A. lobifera*. Moreover, the
*Symbiodinium* symbionts clearly exhibit stress earlier than the host. The different thermal sensitivity of the symbionts and
host of *S. orbiculus* imply that they might be a limiting factor for the host to cope with future warming. A similar apparent
discordance was previously observed in *Amphistegina* by Hallock et al., 2006b which suggested that the ectoplasm of
bleached specimens is "preprogrammed" to continue calcification. Our observation of *S. orbiculus* indicates that this
discordance might be limited to a relatively short time after the bleaching. Possible explanations for the synchronized
response of the *A. lobifera* holobiont in this study are either 1) similar thermal sensitivity of the symbiont and the host or 2)
the weekly resolution of measurements may not capture a short discordance time between the responses of the symbiont and
host.
Our results also reveal different thermal tolerance of the two *S. orbiculus* populations demonstrated by the onset of their
response to 35°C. The main difference between the two populations is the pre-exposure to the very cold winter in the Eastern
Mediterranean compared with much more moderate spring temperatures (Schmidt et al., 2016b; Titelboim et al., 2016). The
substantial effect of seasonal pre-exposure on the thermal tolerance of a population demonstrates the existence of
acclimation processes of the host, the symbiont or both. However, while *A. lobifera* spring population exhibited sensitivity




only to 35°C, a previous study that examined the thermal tolerance of a summer population indicates that the latter
negatively responded already to 30°C (Titelboim et al., 2019). These observations highlight the notion that while pre-
exposure to moderate temperature benefit the holobiont, in cases of extreme temperatures (cold or warm) it might reduce its
thermal tolerance. In the context of ocean warming, this implies that while acclimation may mitigate some increase in SST,
pre-exposure beyond a certain threshold will most likely reduce the thermal tolerance of LBF.
**Conclusions**
Considering the role of LBF in the carbon cycle and as ecosystem engineers, their future with expected warming is a major
concern. Moreover, the relative carbonate production of different LBF species is presently not equal and rising temperatures
will most likely change their relative contribution. Our study emphasizes the role of pre-exposure and acclimation processes
in mitigating the effect of future warming. It implies that with expected rising SST exceeding certain thresholds, pre-
exposure to extreme temperatures will have a negative influence on thermal tolerance. Our study clearly shows that LBF
have different thermal tolerances that are limited by the sensitivity of their eukaryotic algal symbionts. Considering recent
findings on the significant role of the prokaryotic microbiome on the physiological performance of LBF (Prazeres, 2018;
Prazeres et al., 2017), it will be highly valuable also to explore in future studies their specific contribution to the thermal
tolerance of the holobiont .
**Data availability**
All data related to the manuscript is available in Supplement 2.
**Author contribution**
The study was designed by D.T. and D.P. Sampling and culturing experiments were carried out by D.P and D.T. using
facilities provided by S.A.; Interpretation of data and writing of the manuscript were done by all authors: D.P., D.T., and
S.A.
**Competing interests**
The authors declare that they have no conflict of interest.
**Acknowledgment**
We thank Yahel Eshed and Neta Gershon for their assistance in the fieldwork. We acknowledge funding by the Israel
Science Foundation grant No. 941/17, BMBF-MOST cooperation in Marine Sciences grant numbers: 3-15275 and 3-15274.



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
