# Peer review of "Foraminiferal holobiont thermal tolerance under future warming Roommates problems or successful collaboration?"

_Biogeosciences, 2019_

## Referee Comment (RC1) · Anonymous Referee #1 · 14 Feb 2020

This elegant study presents data of a laboratory experiment comparing, for 2 larger symbiont-bearing benthic foraminiferal species, their response to high temperature, in terms of the foraminiferal calcification rate and the photosynthetic rate of their symbionts.

These data are important, interesting, and deserve to be published. The study is very well conceived, and the high quality data are analysed with adequate statistical methods.

The text is rather short, and the information is quite dense. In such a case, the written text should be very precise, and all potential sources of ambiguity should be avoided.

This is not always the case yet.

A main, recurrent problem in the discussion is that systematically, there is confusion between the holobiont (foraminiferal host + symbionts) and the host (the foraminifer without symbionts). Often, the authors speak about the host, when they mean the "whole foraminifer", that is the holobiont. This is not surprising, because it is probably impossible to consider the foraminifer without its symbionts, which represent an essential part of it. For that reason, I think it's impossible to compare the "well-being" of the foraminiferal host with that of the symbionts!

This becomes problematic when the calcification rate alone is supposed to represent perfectly well the general state of the "foraminiferal host". Most times, when the authors compare "the host and the symbionts", in reality, they compare the "calcification rate of the foraminifer" with the (photosynthesis rate of the) symbionts.

I agree that the photosynthetic rate probably describes the health of the symbionts very well, but I am not convinced that the same can be said about the foraminifer and its calcification rate. I would say that many other factors (together) determine (and can inform about) the wellbeing of the foraminiferal holobiont, the wellbeing of its symbionts being one of them!

Summarising, the authors should formulate things more carefully. They compare foraminiferal calcification rates with symbiotic photosynthetic rates. Then that's what they should write!

Similarly, I think that some parts of the discussion go too far. The authors have only tested part of the response of the foraminiferal holobiont to high temperatures. Other indicators (locomotion, feeding, reproduction, etc.) may respond differently, and future climate change will probably lead to changes in other stressors (salinity, oxygenation, carbonate chemistry, etc.) as well. They should therefore be much less affirmative when they discuss the future evolution of larger BF communities.

Detailed comments:

Line 12: a "contribution to thermal tolerance" is somewhat strange. This suggest some pro rata contribution to the tolerance of several factors. In reality I would expect that the overall tolerance is determined by the element which is least tolerant, either the foraminifer or its symbionts.

Line 13: a key question is to what point calcification (for foraminifera) and photosynthetic activity (for symbionts) can be considered representative for their tolerance. For the symbionts, since photosynthesis is a primary life process, this is probably the case. Concerning foraminiferal calcification, it is less evident that this is the best marker of tolerance. I would say overall activity (feeding, locomotion, pseudopod movements) and reproduction are more critical parameters. There are many observations of (active?!) foraminifera under stressed conditions without calcification (even of decalcified forams). Only on the long term, a lack of calcification may lead to disappearance. I think this point should be discussed.

Line 15: "sensitivity to 35°C": what does that mean, if resilience is up to 32°C? The way it is formulated, the authors suggest that there is no sensitivity until 32°C which is evidently wrong. They probably mean something like "progressive loss of life functions between 32°C and 35°C".

Line 16: "future warming will change....). The word "will" is definitely too affirmative. Replace by "may".

Line 17: "a synchronized response": this suggest that there is some deliberate process behind it, like host and symbionts coordinating their activities. Since you don't know this, it is better to use the more neutral tem "synchronous".

Line 20: "pre-exposure to modest temperatures". This is too imprecise. Should be "moderately high temperatures".

Introduction

Lines 28-29: No, it is not this area, but the entire Mediterranean which can be considered as a miniature ocean.

Line 41: "Some species live close to their thermal thresholds": upper or lower? In fact, the further invasion of some LBF (Amphistegina) in the Western Med is hampered by the fact that they are limited by their LOWER temperature threshold (at present, it is too cold for them to go further west). I guess that you mean here that in the Eastern Med, at present, they live close to their UPPER threshold?! Please be more specific.

Lines 43-44: "the relative contribution (positive or negative) of the host and symbiont algae to cope with rising temperatures". As indicated before, this is really strange. The way it is written here, the sentence doesn't make sense. You probably mean "the relative contribution to the tolerance of rising temperature". But also that concept is very strange. This suggests that somehow you can quantify that when a LBF can still function at let's say at 30°C, what proportion of this resistance is due to the foram itself, and what proportion is due to (activities of) the symbionts. I think this is not possible! You simply want to investigate the "relative tolerance of host and symbiont algae to higher temperature". With the underlying idea that the element with the lowest tolerance (host or symbionts) will probably be determinant for the tolerance of the holobiont.

Lines 45-46: "LBF species with different holobiont systems": incorrect formulation: the LBF species IS the holobiont system (combination of host and symbionts)! The 2 species represent 2 different holobiont systems. Methods

Lines 67-68: "however, they were used to produce comparable data to that of related published papers (Schmidt et al., 2016b, 2016a, 2018; Titelboim et al., 2019)."

This sentence is very unclear. "they" should be more specific, like "these light conditions".

But then, also the following part of the sentence doesn't make sense to me. Who used these conditions to produce comparable data? You? Or the cited authors?

[Figure]

But you say the data are comparable to data of these authors. So it is probably your data you talk about?! Then you should write: "however, while using these light conditions, we were able to produce data comparable to those presented in related published papers."

Finally, if I understand you right, I don't see why the fact that you have comparable data as other others with the same conditions shows that these conditions are ok?! Maybe both your and other studies have unreliable results because by using insufficient light, you may have added an additional stress factor?! This eventuality should be discussed!

2.2 Laboratory manipulative experiments

Line 77: "acclimated under constant conditions": it is essential so say at what temperature!

2.3. Results

Fig. 2b: no data for photosynthetic activity of S. orbiculus in winter. Why not? Explain in methods section! However, the caption of Suppl. Table S.3.2. mentions 3 groups! Lines 115-119. "in the winter population, calcification decreases already after one week and is inhibited after three weeks"

This looks like an over-interpretation to me: in view of the overlapping error bars, I don't think that the "week 1" values are statistically different for 25°C and 35°C! The supplementary table doesn't inform us about this.

Fig. 3: I'm intrigued by the last line: "Abnormal measurement is marked as extreme and is not calculated as part of the average and error." I would write "a single abnormal measurement, obtained after x weeks. . ...". You have to add the info in which week this measurement was made!

The regular text also describes this anomalous measurement and says ". . .as it is clearly damaged from sample handling". I don't see how inadequate sample handling can lead to such a value! I would simply not explain this single anomalous value.

Chapter 3.2. Amphistegina

Line 131 "Both calcification and photosynthesis responses remain synchronized throughout the experiment".

I don't think you can say that. "Remain synchronized" means that there is an intrinsic interaction mechanism which explains why the responses of these two parameters are synchronous. "synchronised" is a wrong word. You should write: "are synchronous throughout the experiment".

Line 133: "calcification and photosynthesis were both inhibited".

However, calcification values are still slightly positive, so calcification doesn't seem to be (entirely) inhibited!

Line 133 "and net photosynthesis was negative"

That doesn't add anything to "inhibited photosynthesis". If you want to mention this, it should come BEFORE the conclusion of inhibited photosynthesis.

Discussion

Lines 143-145: "Specifically, our results predict that with rising temperatures the relative contribution of S. orbiculus will increase since its calcification is not inhibited even at extreme temperatures, contrary to A. lobifera".

I have three problems with this sentence:

1) I would prefer when, before jumping to such a conclusion, you first briefly summarise the differences you found between the two species.

2) Next, as said before, to me, the situation doesn't seem so "black-white" as you suggest (inhibition – no inhibition): for both species the calcification rate goes down at 35°C. It is true that the values go down much more for A. lobifera, but it doesn't become zero. If you think that a value of 10 $\mu$Mol carbonate per individual per week means "no

calcification", then you have to explain why!

3) I think this conclusion goes much farther than you can go with your present results. I could imagine that a species no longer calcifies in the warmest month, but still survives these months without any major problems. Your observations only suggest that A. lobifera resists less well than S. orbicularis to high temperature. But that's not enough to go as far as you go, by concluding that in future, Sorites will progressively replace Amphistegina. In fact, temperature is one stress factor, but there may be others, which could covary with temperature, like salinity. Maybe the tolerance of the two species to raised salinity (or any other stress factor) is exactly the opposite?!

Lines 159-166: here the authors summarise their main results. But this is way too late. This paragraph should already be inserted at line 143/144, before presenting the overly speculative final conclusion/suggestion.

Line 164 again mentions "inhibition of calcification", whereas the measure values are not zero. A more "nuanced" wording is absolutely necessary!

Line 166: "Moreover, the Symbiodinium symbionts clearly exhibit stress earlier than the host." True, that is to say, for the indicator you use, i.e., calcification rate. However, this may not be the best indicator. Maybe the host would show stress just as early (or even earlier) if you would use another indicator (e.g., locomotion, feeding behaviour, reproduction, etc.). And finally, with symbionts showing signs of stress, it is hard to image the foram itself is not "feeling" signs of stress!

I simply want to underline that in my opinion you can't reduce the "well-being" of the foram to its calcification efficiency. This is only one element out of many others, which may not even be critical!

Lines 166-67: "The different thermal sensitivity of the symbionts and host of S. orbiculus".

Same remark here. You can't base your ideas on the "thermal sensitivity of the host" (=

the whole holobiont) only on its calcification rate. I would say that the thermal sensitivity of S. orbiculus depends both on the thermal sensitivity of its calcification rate, on the thermal sensitivity of its symbionts and on the thermal sensitivity of many other of its life processes.

I think you should rather write: "The different thermal sensitivity of the calcification rate and of the symbionts of S. orbiculus".

Line 168-69: "Hallock et al., 2006b which (=who) suggested that the ectoplasm of bleached specimens is "preprogrammed" to continue calcification." → This sentence needs some more explanation!

Lines 169-70: "Our observation of S. orbiculus indicates that this discordance might be limited to a relatively short time after the bleaching". → I have no idea what you are talking about! What "discordance" do you mean? (probably the wrong word!). What can your observations on S. orbicularis tell us about bleaching? I'm lost! Please clarify!

Conclusion

Lines 187-88 "Our study emphasizes the role of pre-exposure and acclimation processes in mitigating the effect of future warming." It is very strange to me that this point, which is only discussed very briefly at the end of the discussion, suddenly becomes the main conclusion of your work!

---

## Referee Comment (RC2) · Anonymous Referee #2 · 25 Feb 2020

Dear authors,

I do agree with reviewer 1 in many of the issues raised, but have not had the time yet to write my detailed review report. In order to facilitate an efficient review process, I guess it would be more useful if I comment on the updated version of the manuscript, or should I write a report on the first draft, although the authors already addressed some of the issues? If you would like me to review the newer manuscript, please provide a PDF of the updated version that includes the changes made. So far, the PDF available seems to be the former version.

---

## Editor Comment (EC1) · Jack Middelburg (Editor) · 25 Feb 2020

Dear Referee:

Thank you for accepting to evaluate this paper. The usual procedure at BG (and most other copernicus journals) is to have formal reviews and responses of authors before the associate editor decides whether the authors are invited to submit a revised version. Consequently, referees may identify similar shortcoming or provide similar suggestions. It is not the idea that the authors revise after that one review is in and then the next referee evaluates the paper. (This could be a model, but would cause much delay if someone in this chain is not responding).

[Figure]

I would therefore very much appreciate if you would provide feed-back/comments/suggestions to the author at this stage. I anticipate that the revised version may need another round of review.

With best regards,

Jack Middelburg, handling associate editor

---

## Referee Comment (RC3) · Anonymous Referee #2 · 26 Feb 2020

General comments:

The study presented by Pinko et al. represents a comparative study of two different common LBF species, with different shell and symbiont types, exposed to elevated temperature over few weeks. The two main proxies assessed give insights into photosymbiont performance and holobiont health. Due to subtle differences, the authors conclude that Sorites orbiculus will be less affected by climate change than Amphistegina lobifera. They also claim insights into distinct effects of pre-exposure to moderate temperatures regarding the LBFs thermal tolerance. Along the lines of former studies, the experiment shows that there are species-specific thresholds regarding tempera-

ture and duration of exposure, and that LBF from the Eastern Mediterranean, which are most likely Lessepsian invaders from the Red Sea, have a relatively high thermal tolerance. The further confirm that the photosymbionts seem to be the 'weaker' member of this symbiotic association, showing the earlier stress response. Hence, the study give further important proof of prior hypotheses on LBF thermal stress responses, and adds to the knowledge of species-dependent thresholds. Hence, I consider it important and valid to publish this data. However, the novel insights are limited, as I do not think that calcification can be considered as a host-specific response (as they suggest), and therefore this study does not assess the relative contribution of host and symbionts (see specific comments).

Specific comments:

-Calcification cannot be considered as a host proxy, as it is largely influenced by photosynthesis. It is hence, as in many other studies, a holobiont proxy. Although prior studies mostly used possibly less precise methods to assess growth (e.g. increase in surface area or buoyant weight in studies by Schmidt, Prazeres, Stuhr, Hallock and others), it basically gives similar information. Nearly all studies on LBF stress response assessed at least one holobiont parameter such as growth (often also others to get a better picture, as calcification / growth can be limited due to other factors that are independent of stress, hence, it is not a very good parameter anyways), and one or more photosymbiont parameters. The only study to my knowledge that actually managed to gain host-specific insights was Stuhr et al. 2018 (Scientific Reports) by differentiating between host and symbionts on the protein level. But even here, the influence of photosymbionts stress on host stress cannot be fully excluded.

- Due to the lack of novelty described above, I would suggest to the authors to focus more on the comparison between A. lobifera and S. orbiculus, and the detected differences in time-related responses (seasons and experimental duration), and emphasize these in more detail.

[Figure]

- Furthermore, the methods section is very short and lacking a lot of details, descriptions etc., and many crucial information has unfortunately been moved to the supplementary materials. The same applies to some of the results, e.g. the statistics, which should be at least indicated in the text or the figures where significant. Even with the results provided in the supplementary materials, it is not possible to fully judge where statistically significant variation were detected due to the poor representation and lacking explanations.

- I am also wondering why no further parameters were tested, and calcification and photosynthesis normalized by individual, which is very unusual and prohibits comparison with other studies. It is also less precise if not all individual were of the same size (which they most likely were not). And I wonder, if you measured photosynthetic activity via oxygen production, why didn't you simply also measure respiration via oxygen consumption in darkness? This would have provided another valuable indicator for holobiont condition, and would have allowed to calculate gross oxygen production. So far, you only provide net values for photosynthesis, not considering that respiration is likely to be higher under high temperatures, which results in lower net photosynthesis, even though the actual production of oxygen may be constant.

- Lastly, I think this study could further be improved by discussing some interesting observational details such as bleaching (Did you observe it? In which species? Mottled or more gradual? Or mortality? Reproduction?) as well as by including time into the statistical evaluation of the results (e.g. two-factorial analysis of variance with time and temperature as factors).

Technical comments:

L2: "Roommate problems or successful collaboration?" this title sounds catchy, but the question is still as valid as before... I don't think the paper is resolving this question, and also the first part of the title is very broad (e.g. climate change includes more than just temperature stress) and should be a bit more specific

L11: "... hyaline diatom-bearing Amphistegina lobifera and the proceallaneaous dinoflagellate-bearing Sorites ..."

L12-13: see discussion above

L16-L17: "future warming will significantly shift the relative contribution..." this is taking the implications way too far. You only see small differences in their response to 35°C in respect of timing. Please be more specific (in general) and stick to what you actually show.

L18-21: You mention pre-exposure for the first time here, and it is rarely described in the manuscript in general. What do you mean by this? The season? Or different acclimation temperatures? I also don't understand why you suggest that it reduces thermal tolerance. Please reconsider these statements and adjust to your results and the discussion.

L27-28: "... one of the regions most affected..."

L30: "Symbiont-bearing large benthic foraminifera ..."

L34-35: To my knowledge, none of these studies really provided evidence for temperature control on symbionts composition. Some suggested that there may be a connection, but statements saying that they are "strongly controlled" would definitely require further proof, especially since other studies did show extremely flexible relationships (e.g. several Lee et al. studies, Pochon et al. 2007, Schmidt et al. 2016)

L45: delete "calcifiers"

L55-56: there are much earlier studies that describe these species in much more detail (such as Hansen & Burchard 1977 and Hottinger 1977) that deserve to be cited here.

L59: "... Israel, during..."

L60: the picked size fraction is crucial when it comes to assessing growth/calcification as it is strongly linked to ontogenetic phase. Hence, please provide this information in

the manuscript and not in the supplemental materials.

L62: the same accounts for the sample sizes. It is important to know in order to judge the power of the study. And "... 60-ml airtight..."

L 68: there is still a lot of detail missing regarding the culture conditions: was there water flow? What kind of water were they in? Were they fed? pH? ...

L65: Which temperatures? In the baths or the flasks? What means regularly? Once a week or once per hour would both be regularly but are very different...

L71: Why did you use calcification rate as the only parameter? Please explain.

L74: Same for: why did you only include A. lobifera for the spring experiment?

L77: Which "constant conditions"?

L79-80: How many samples didn't show oxygen production? And any suggestion why?

L81: How was temperature adjusted? What instruments did you use to control this?

L82: where are these temperatures expected? In the Med Sea? The Red Sea?

L85: Was the water filtered? Pre-conditioned temperature-wise?

L85-90: Please give more details on the method. Did you do this in the 60-ml flasks? What was the time frame? Were foraminifera pooled?

L91-93: Same details are missing here, as well as references. What instruments did you use? What light? Which temperatures? "... $\mu$g L-1 ..." and please define RDO. What was the accuracy value? And in general, you normalize both parameters by specimen. For better comparability, they are usually normalized by species size (given by surface area or weight etc.). Please consider doing so.

L95-97: Please give n for each parameter, treatment, time point, described which data got transformed, and which tests exactly got chosen "accordingly".

L99: " . . .cases where normality . . . non-parametric Kruskal-Wallis test . . ."

L100: Please name the "proper" post-hoc tests.

L106: you cannot say whether there would be differences in Al. lobifera between winter and spring, but sounds as if you do. Please rephrase.

L108: What is the "x" in your box plots? Please indicate significant differences by a letter report. And give n in captions. Also provide full species names and specify what the whiskers represent (SE?).

L112&L130: keep descriptions consistent

L113f: Please avoid expressions like "much higher", "a decrease" or "substantially different". What does that mean? Please provide statistical test results and/or how big is the difference (twice as high, ~20 %...)

L120: "The symbionts' photosynthetic . . ." and how is the "sensitivity pattern" different? Apart from one week in the 35°C treatment, they look very similar to me.

L122: As mentioned before, please provide overview of statistical results here, ideally in figure.

L123: If you mention this "abnormal value" please state in which way it was abnormal and why you suspect this to be related to handling. You say it's not used for average and SE calculations. Does that also mean the further statistical analyses?

Fig. 3: please jitter weeks more, so they are easier to recognize. In which week did you have the extreme value?

L128: "significant negative response" I am not sure the word significant is used correctly here.

L134: "week" and what means "massive bleaching"? Please describe, give proportions etc.

[Figure]

L135-136: "... between 25°C and 32°C, and was thus clearly ..." As described in the general comments, please describe it more. What may be the reason for this bleaching? Were both species affected? A. lobifera could have a lower light tolerance than Sorites and could therefore bleach. Or since it only affected the lower temperature range, it could actually be related to reproduction. Why didn't you exclude the bleached specimens from the analyses? I suggest to include the data anyways, e.g. only for comparison in the supplementary, or to conduct analyses on those samples that had no bleaching. Moreover, as the 35°C had no bleaching, please provide at least these values to compare with former weeks.

L142-143: Well, I think "clear differences in thermal tolerance" is a bit exaggerated. I would call them rather subtle.

L144: This is not true, the calcification seems lower at 35°C (significances missing), at least in the winter populations. Plus, the experimental exposure in spring was shorter, so the response may have just been delayed, as suggested by the reduced oxygen production.

L147: please rephrase, something here doesn't make sense. And I again don't agree that you can state that there is a "strong dependence", as many other factors have been shown to be at least as important. Please also include some newer references here.

L149&L161: "dinoflagellates" and change to "Symbiodinium" to Symbiodiniaceae (as you seem to be aware this taxonomic system has been revised)

L153: "... control a holobionts ..."

L154-155: What "mechanism to cope with thermal stress was observed"? Please describe, I do not think this paper actually showed 'shuffling'.

L156: "... explain the observation ..."

L159: "... describes ..."

L164: "... 35°C, whereas in ..."

L165: "... inhibited, indicating that it is..."

L166: That means you refer to Sorites only? When is "earlier"?

L168: An apparent higher sensitivity (earlier/stronger response of the symbionts than the holobiont) was also observed by other studies such as Prazeres et al. 2017, Stuhr et al. 2017, Schmidt et al. 2016...

L170: Here you mention bleaching again: so were these specimens that you measured calcification on already bleached? If there was, this may also indicate that there was another stress factor such as too high light intensity or the wrong light spectrum (e.g. Hallock once showed that blue light facilitated high growth rates but at the same time led to bleaching). Please discuss your observations.

L172: The resolution is a very important point! One week is a long time for a foraminifera! Plus, so far I don't even know how long your calcification measurements or photosynthesis measurements took. They don't calcify continuously all day long, so the time frame may strongly influence the results. The same accounts for the photosynthesis, which varies over the time of the day.

L174: Please specify the time of the onset.

L175: What do you mean by "very cold"? I think that is very relative... give a temperature range of what is usually encountered in the Med Sea in winter and spring, and ideally state what were the temperature measured during sampling in the methods section.

L178: "... symbionts, or both. However, while the A. lobifera spring ... "

L180: In which way did they respond "negatively"? Please be a bit more specific so the reader does not have to go back to each of the studies you cite to find you what you mean, and "... that, while ..."

[Figure]

L181: "... temperatures benefits the ..."

L186-187: There have been studies modeling the future changes in distribution and hence contribution, which should be cited here (e.g. Weinmann et al. 2013, 2017)

L187: If you mention pre-exposure here (as in the abstract) please elaborate a bit more in which way they had different pre-exposures in the methods as well as the discussion parts.

L188-189: I don't understand this statement at all. Why? Where is you evidence for that?

L189: Again, I find "clearly shows" a bit exaggerated. Supplement 1: Is the number of replicates given the value before or after exclusion of same samples? Why are the numbers different for the different time points? How did you deal with this unbalanced design in your statistical analyses?

You used once filtered and once unfiltered water. Why? And why did you pre-condition the spring Sorites to another temperature than the rest? This is very crucial information and must not be excluded from the actual manuscript! I am not sure if you can compare your data the way you do with all these differences.

Sometimes you give two numbers after the comma, sometimes six or other... please be consistent (and usually its three).

Why do you give four stars (they are actually called asterisk)? Usually, these are used to indicate the level of significance, from one (lower end of significance) to three (highly significant).

Why is some text red? Please explain in captions.

Table S2.2 and others: What are "1, 2 and 3" in your column headers? I cannot understand your statistical results if I don't know what is which group.

Put spacing equally before and after "=", but not after "(".

Table S5.2: "Stars indicate homogenous groups and thus significant differences between them"? This makes no sense to me, because if they are homogenous, they are similar, so no difference...

---

## Author Response (AR1)

אוניברסיטת בן-גוריון בנגב
Ben-Gurion University of the Negev

Department of Geological and Environmental Sciences

March 16, 2020

Dear Jack Middelburg,

Please find enclosed the revised version of our manuscript that I resubmit in the name of all co-authors. We believe that the reviewers raised some important issues that we have considered and changed the manuscript accordingly. The main issues raised by the reviewers were:

1) The use of our approach to separately examine the well-being of the foraminiferal host and the symbionts. We maintain that our approach is valid since calcification is a physiological trait performed only by the foraminifera and thus present a direct proxy of its wellbeing. The same is valid for photosynthesis, which is a physiological trait of the algal symbionts. Because of the exclusiveness of each parameter, we have selected them in order to get specific indications for the two components of the holobiont. Stress of the symbionts will indeed affect the host and vice versa. However, we are not examining the interaction between them but the specific response of foraminifera and algae. This is explained in more details including examples from previous publications in the responses to each reviewer and the explanation is also added to the manuscript. Additionally, to further clarify this issue in the manuscript, we have rephrased throughout the text to indicate "foraminiferal calcification rates and symbiotic net photosynthesis" as suggested by reviewer #1.

2) 2) Not enough strong evidence to support the discussion regarding the influence of pre-exposure on the thermal sensitivity of the holobionts. We accept the criticism that there are differences between the experiments other than temperature and that to make this case this will have to be done separately in a much more comprehensive manner. Therefore, we have excluded this part from the manuscript.

All other comments have been followed and are implemented in the current version of the MS. We believe that the result is a much-improved manuscript. Below is a point-by-point reply to all comments made by the reviewers (in blue).

Kind regards,

Dr Danna Titelboim

**Response to comments by Reviewer 1:**

This elegant study presents data of a laboratory experiment comparing, for 2 larger symbiont-bearing benthic foraminiferal species, their response to high temperature, in terms of the foraminiferal calcification rate and the photosynthetic rate of their symbionts.

These data are important, interesting, and deserve to be published. The study is very well conceived, and the high quality data are analysed with adequate statistical methods.

The text is rather short, and the information is quite dense. In such a case, the written text should be very precise, and all potential sources of ambiguity should be avoided. This is not always the case yet. A main, recurrent problem in the discussion is that systematically, there is confusion between the holobiont (foraminiferal host + symbionts) and the host (the foraminifer without symbionts). Often, the authors speak about the host, when they mean the "whole foraminifer", that is the holobiont. This is not surprising, because it is probably impossible to consider the foraminifer without its symbionts, which represent an essential part of it. For that reason, I think it's impossible to compare the "well-being" of the foraminiferal host with that of the symbionts!

This becomes problematic when the calcification rate alone is supposed to represent perfectly well the general state of the "foraminiferal host". Most times, when the authors compare "the host and the symbionts", in reality, they compare the "calcification rate of the foraminifer" with the (photosynthesis rate of the) symbionts.

I agree that the photosynthetic rate probably describes the health of the symbionts very well, but I am not convinced that the same can be said about the foraminifer and its calcification rate. I would say that many other factors (together) determine (and can inform about) the wellbeing of the foraminiferal holobiont, the wellbeing of its symbionts being one of them!

Summarising, the authors should formulate things more carefully. They compare foraminiferal calcification rates with symbiotic photosynthetic rates. Then that's what they should write!

Response: See response to main issue 1 in the first part of the letter. We have rephrased throughout the text to indicate "foraminiferal calcification rates and symbiotic net photosynthesis" as suggested by the reviewer.

Similarly, I think that some parts of the discussion go too far. The authors have only tested part of the response of the foraminiferal holobiont to high temperatures. Other indicators (locomotion, feeding, reproduction, etc.) may respond differently, and future climate change will probably lead to changes in other stressors (salinity, oxygenation, carbonate chemistry, etc.) as well. They should therefore be much less affirmative when they discuss the future evolution of larger BF communities.

Response: We agree with this point and changed it throughout the text. We specifically address the issues of other well-being indicators and of other stressors in the detailed comments below (comment 2 and 23).

Detailed comments:

1. Line 12: a "contribution to thermal tolerance" is somewhat strange. This suggest some pro rata contribution to the tolerance of several factors. In reality I would expect that the overall tolerance is determined by the element which is least tolerant, either the foraminifer or its symbionts.

Changed to: "In order to assess the holobiont thermal tolerance we separately evaluated foraminiferal calcification rates with symbiotic photosynthetic rates"

2. Line 13: a key question is to what point calcification (for foraminifera) and photosynthetic activity (for symbionts) can be considered representative for their tolerance. For the symbionts, since photosynthesis is a primary life process, this is probably the case. Concerning foraminiferal calcification, it is less evident that this is the best marker of tolerance. I would say overall activity (feeding, locomotion, pseudopod movements) and reproduction are more critical parameters. There are many observations of (active?!) foraminifera under stressed conditions without calcification (even of decalcified forams). Only on the long term, a lack of calcification may lead to disappearance. I think this point should be discussed.

Previous studies have demonstrated that calcification rates can be used as a direct parameter for comparing the temperature sensitivity of different calcifying organisms, this is due to the fact that calcification involves a profound consumption of energy. Therefore, calcification rates are directly linked to the range of optimal to suboptimal conditions of the organism (Lough & Barnes, 2000; Carricart-Ganivet et al., 2012). This was specifically demonstrated on different species of foraminifera (Schmidt et al., 2011, 2015, 2016a, 2016b; Vogel and Uthicke ,2012, Uthicke and Fabricius, 2012, Evans et al., 2015).

The Alkalinity anomaly method present an ideal experimental approach for detecting even subtle differences in performance under different treatments and while it is true that the overall activity and reproduction are critical parameters to indicate the well-being of foraminifera they will not identify as clearly and as quantitively the small differences between treatments. We added a short explanation of this in the method section.

3. Line 15: "sensitivity to 35°C": what does that mean, if resilience is up to 32°C? The way it is formulated, the authors suggest that there is no sensitivity until 32°C which is evidently wrong. They probably mean something like "progressive loss of life functions between 32°C and 35°C".

Changes as suggested.

4. Line 16: "future warming will change. . ..). The word "will" is definitely too affirmative. Replace by "may".

Changes as suggested.

5. Line 17: "a synchronized response": this suggest that there is some deliberate process behind it, like host and symbionts coordinating their activities. Since you don't know this, it is better to use the more neutral term "synchronous".

Changes as suggested.

6. Line 20: "pre-exposure to modest temperatures". This is too imprecise. Should be "moderately high temperatures".

The discussion regarding pre-exposure has been excluded from the manuscript (see response to the main issue #2 in the first part of the letter).

Introduction

7. Lines28-29: No, it is not this area, but the entire Mediterranean which can be considered as a miniature ocean.

Changes as suggested.

8. Line 41: "Some species live close to their thermal thresholds": upper or lower? In fact, the further invasion of some LBF (*Amphistegina*) in the Western Med is hampered by the fact that they are limited by their LOWER temperature threshold (at present, it is too cold for them to go further west). I guess that you mean here that in the Eastern Med, at present, they live close to their UPPER threshold?! Please be more specific.

The text is corrected to indicate upper thresholds. It is true that species are limited by their lower temperature threshold, but it is still important to consider upper threshold as even small increase in temperature, predicted in the relatively near future will influence species that are close to their upper thresholds.

9. Lines43-44: "the relative contribution (positive or negative) of the host and symbiont algae to cope with rising temperatures". As indicated before, this is really strange. The way it is written here, the sentence doesn't make sense. You probably mean "the relative contribution to the tolerance of rising temperature". But also that concept is very strange. This suggests that somehow you can quantify that when a LBF can still function at let's say at 30°C, what proportion of this resistance is due to the foram itself, and what proportion is due to (activities of) the symbionts. I think this is not possible! You simply want to investigate the "relative tolerance of host and symbiont algae to higher temperature". With the underlying idea that the element with the lowest tolerance (host or symbionts) will probably be determinant for the tolerance of the holobiont.

Indeed, we mean that the component with lowest tolerance will limit the tolerance of the holobiont. We changed the wording to make it clearer.

10. Lines 45-46: "LBF species with different holobiont systems": incorrect formulation: the LBF species IS the holobiont system (combination of host and symbionts)! The 2 species represent 2 different holobiont systems.

Changed.

Methods

11. Lines 67-68: "however, they were used to produce comparable data to that of related published papers (Schmidt et al., 2016b, 2016a, 2018; Titelboim et al., 2019)." This sentence is very unclear. "they" should be more specific, like "these light conditions". But then, also the following part of the sentence doesn't make sense to me. Who used these conditions to produce comparable data? You? Or the cited authors? But you say the data are comparable to data of these authors. So it is probably your data you talk about?! Then you should write: "however, while using these light conditions, we were able to produce data comparable to those presented in related published papers."

Changed.

12. Finally, if I understand you right, I don't see why the fact that you have comparable data as other others with the same conditions shows that these conditions are ok?! Maybe both your and other studies have unreliable results because by using insufficient light, you may have added an additional stress factor?! This eventuality should be discussed!

Ziegler and Uthicke, 2011 specifically indicate that photosymbionts of LBF acclimate very rapidly to different light leves in under 48 hours. This means that our 10 days acclimation is sufficient for them to adjust to the specific light level provided during the experiment. Furthermore, it is important to note that the light level is low in respect to the photosynthetic optimum and not other physiological functions of the holobiont. Based on our own experience of culturing *Amphistegina* in the lab for several years we can confirm that specimens look healthy (colorized, strong motility), and show substantial growth by producing new chambers in similar manner as field specimens.

2.2 Laboratory manipulative experiments

13. Line 77: "acclimated under constant conditions": it is essential so say at what temperature!

This was all described in the supplementary but is now moved to the main text in the revised manuscript.

2.3. Results

14. Fig. 2b: no data for photosynthetic activity of *S. orbiculus* in winter. Why not? Explain in methods section! However, the caption of Suppl. Table S.3.2. mentions 3 groups!

This part has been excluded from the manuscript (see response to the main issue #2 in the first part of the letter).

15. Lines 115-119. "in the winter population, calcification decreases already after one week and is inhibited after three weeks". This looks like an over-interpretation to me: in view of the overlapping error bars, I don't think that the "week 1" values are statistically different for 25°C and 35°C! The supplementary table doesn't inform us about this.

This part has been excluded from the manuscript (see response to the main issue #2 in the first part of the letter).

16. Fig. 3: I'm intrigued by the last line: "Abnormal measurement is marked as extreme and is not calculated as part of the average and error." I would write "a single abnormal measurement, obtained after x weeks. . ...". You have to add the info in which week this measurement was made!

Added to the main text and to the caption.

17. The regular text also describes this anomalous measurement and says "...as it is clearly damaged from sample handling". I don't see how inadequate sample handling can lead to such a value! I would simply not explain this single anomalous value.

This part is deleted.

Chapter 3.2. *Amphistegina*

18. Line 131 "Both calcification and photosynthesis responses remain synchronized throughout the experiment". I don't think you can say that. "Remain synchronized" means that there is an intrinsic interaction mechanism which explains why the responses of these two parameters are synchronous. "synchronised" is a wrong word. You should write: "are synchronous throughout the experiment".

Changes as suggested.

19. Line 133: "calcification and photosynthesis were both inhibited". However, calcification values are still slightly positive, so calcification doesn't seem to be (entirely) inhibited!

Inhibited is replaced with "severely reduced"

20. Line 133 "and net photosynthesis was negative" That doesn't add anything to "inhibited photosynthesis". If you want to mention this, it should come BEFORE the conclusion of inhibited photosynthesis.

Since the first part of the sentence is now changed from "inhibited" to "severely reduced", it does add information about the result of this reduction.

Discussion

Lines 143-145: "Specifically, our results predict that with rising temperatures the relative contribution of *S. orbiculus* will increase since its calcification is not inhibited even at extreme temperatures, contrary to *A. lobifera*".

I have three problems with this sentence:

21. (1) I would prefer when, before jumping to such a conclusion, you first briefly summarise the differences you found between the two species.

We moved the paragraph summarizing the differences between the species (this also refers to comment 24).

22. (2) Next, as said before, to me, the situation doesn't seem so "black-white" as you suggest (inhibition – no inhibition): for both species the calcification rate goes down at 35°C. It is true that the values go down much more for *A. lobifera*, but it doesn't become zero. If you think that a value of 10 µMol carbonate per individual per week means "no calcification", then you have to explain why!

Calcification rate values of 10 µMol carbonate is within the precision of the alkalinity measurements in this study. In fact, ±10 µMol is the maximum error of our results. However, we accept this comment since values might be slightly higher than zero and rephrased to - near inhibition.

23. (3) I think this conclusion goes much farther than you can go with your present results. I could imagine that a species no longer calcifies in the warmest month, but still survives these months without any major problems. Your observations only suggest that *A. lobifera* resists less well than *S. orbicularis* to high temperature. But that's not enough to go as far as you go, by concluding that in future, *Sorites* will progressively replace *Amphistegina*. In fact, temperature is one stress factor, but there may be others, which could covary with temperature, like salinity. Maybe the tolerance of the two species to raised salinity (or any other stress factor) is exactly the opposite?

We have adjusted the conclusions to better represent the scope and the significance of this study. Specifically, concerning the possible effect of salinity, our previous studies indicate that temperature is a much more prominent stressor than salinity (Titelboim et al., 2016, Kenigsberg et al., 2020). This is further supported by culturing experiment that is presently being carried out in our laboratory which is testing the separate and combined response of *A. lobifera* to elevated temperature and salinity. However, since this point is also true for other stressors and changes caused by future climate change, we have rephrased the conclusion.

24. Lines 159-166: there the authors summarise their main results. But this is way too late. This paragraph should already be inserted at line 143/144, before presenting the overly speculative final conclusion/suggestion.

We agree with this comment and moved this part to the beginning of the discussion.

25. Line 164 again mentions "inhibition of calcification", whereas the measure values are not zero. A more "nuanced" wording is absolutely necessary!

We agree, see response to comment 19 and 22 (inhibited is replaced with "severely reduced")

26. Line 166: "Moreover, the Symbiodinium symbionts clearly exhibit stress earlier than the host." True, that is to say, for the indicator you use, i.e., calcification rate. However, this may not be the best indicator. Maybe the host would show stress just as early (or even earlier) if you would use another indicator (e.g., locomotion, feeding behaviour, reproduction, etc.). And finally, with symbionts showing signs of stress, it is hard to image the foram itself is not "feeling" signs of stress!

I simply want to underline that in my opinion you can't reduce the "well- being" of the foram to its calcification efficiency. This is only one element out of many others, which may not even be critical!

See response to comment 2.

27. Lines 166-67: "The different thermal sensitivity of the symbionts and host of *S. orbiculus*". Same remark here. You can't base your ideas on the "thermal sensitivity of the host" (= the whole holobiont) only on its calcification rate. I would say that the thermal sensitivity of *S. orbiculus* depends both on the thermal sensitivity of its calcification rate, on the thermal sensitivity of its symbionts and on the thermal sensitivity of many other of its life processes.

I think you should rather write: "The different thermal sensitivity of the calcification rate and of the symbionts of *S. orbiculus*".

Agree. We have changed this sentence as suggested.

28. Line 168-69: "Hallock et al., 2006b which (=who) suggested that the ectoplasm of bleached specimens is "preprogrammed" to continue calcification." → This sentence needs some more explanation!

This notion was given by Hallock et al., 2006b to explain their own observations on bleaching in *Amphistegina*.

29. Lines 169-70: "Our observation of *S. orbiculus* indicates that this discordance might be limited to a relatively short time after the bleaching". → I have no idea what you are talking about! What "discordance" do you mean? (probably the wrong word!). What can your observations on *S. orbicularis* tell us about bleaching? I'm lost! Please clarify!

We removed these sentences.

Conclusion

30. Lines 187-88 "Our study emphasizes the role of pre-exposure and acclimation processes in mitigating the effect of future warming." It is very strange to me that this point, which is only discussed very briefly at the end of the discussion, suddenly be- comes the main conclusion of your work!

The discussion regarding pre-exposure has been excluded from the manuscript (see response to the main issue #2 in the first part of the letter).

**Response to comments by Reviewer 2:**

General comments:

The study presented by Pinko et al. represents a comparative study of two different common LBF species, with different shell and symbiont types, exposed to elevated temperature over few weeks. The two main proxies assessed give insights into photosymbiont performance and holobiont health. Due to subtle differences, the authors conclude that *Sorites orbiculus* will be less affected by climate change than *Amphistegina lobifera*. They also claim insights into distinct effects of preexposure to moderate temperatures regarding the LBFs thermal tolerance. Along the lines of former studies, the experiment shows that there are species-specific thresholds regarding temperature and duration of exposure, and that LBF from the Eastern Mediterranean, which are most likely Lessepsian invaders from the Red Sea, have a relatively high thermal tolerance. The further confirm that the photosymbionts seem to be the 'weaker' member of this symbiotic association, showing the earlier stress response. Hence, the study give further important proof of prior hypotheses on LBF thermal stress responses, and adds to the knowledge of species- dependent thresholds. Hence, I consider it important and valid to publish this data. However, the novel insights are limited, as I do not think that calcification can be considered as a host-specific response (as they suggest), and therefore this study does not assess the relative contribution of host and symbionts (see specific comments).

Specific comments:

1) Calcification cannot be considered as a host proxy, as it is largely influenced by photosynthesis. It is hence, as in many other studies, a holobiont proxy.

It is true that many key elements are involved in the biological machinery of the calcification process. Among these is the mutualistic partnership between the algal symbionts and the foraminiferal host. Nevertheless, calcification is a physiological trait preformed only by the foraminifera and thus present a direct proxy of its wellbeing (as in other calcifying organisms). This is based on a common observation that stress lowers the physiological activities that involves high consumption of energy. The same is true for photosynthesis, which is a physiological trait of the algal symbionts. Because of the exclusiveness of each parameter we have selected them to disentangle the complex relationship between the two components of the holobiont.

Since this issue was raised by both reviewers, we recognize the need for clarification in the paper and added an explanation of the rational to the text at the end of the introduction.

Although prior studies mostly used possibly less precise methods to assess growth (e.g. increase in surface area or buoyant weight in studies by Schmidt, Prazeres, Stuhr, Hallock and others), it basically gives similar information.

The Alkalinity anomaly method provides similar information to that produced by the measure of increase in surface area or buoyant weight. However, its main advantage is that it is much more accurate and thus presents an ideal experimental approach for detecting even subtle differences in performance under different treatments. This technique requires high level of analytical expertise and meticulous work measuring large number of samples (in order to replicate properly) but it can detect differences of even few single micromolars in carbonate production. Thus, this method is highly beneficial over the other common method.

Nearly all studies on LBF stress response assessed at least one holobiont parameter such as growth (often also others to get a better picture, as calcification / growth can be limited due to other factors that are independent of stress, hence, it is not a very good parameter anyways), and one or more photosymbiont parameters. The only study to my knowledge that actually managed to gain host-specific insights was Stuhr et al. 2018 (Scientific Reports) by differentiating between host and symbionts on the protein level. But even here, the influence of photosymbionts stress on host stress cannot be fully excluded.

It is true that stress of the symbiont will affect the host and vice versa. However, it is important to try and find indicators related to each of the component. Stuhr et al. 2018 identified differential expression in protein related specifically to host or symbiont. Under the same logic in this study we examine physiological activity only related to one of the components (calcification of foraminifera and photosynthesis of symbiont algae). The reviewer indicates that even in the data from Stuhr et al. 2018 the host is affected by symbionts stress but still doesn't exclude the fact that the stress is experienced by the foraminifera, we believe that the case is similar with our approach.

Due to the lack of novelty described above, I would suggest to the authors to focus more on the comparison between *A. lobifera* and *S. orbiculus*, and the detected differences in time-related responses (seasons and experimental duration), and emphasize these in more detail.

We changed the discussion to focus more on the comparison of the differences between the holobionts.

Furthermore, the methods section is very short and lacking a lot of details, descriptions etc., and many crucial information has unfortunately been moved to the supplementary materials. The same applies to some of the results, e.g. the statistics, which should be at least indicated in the text or the figures where significant. Even with the results provided in the supplementary materials, it is not possible to fully judge where statistically significant variation were detected due to the poor representation and lacking explanations.

We have added the information to the methods and results sections and the statistical indication of significance (test and p value) throughout the manuscript

I am also wondering why no further parameters were tested, and calcification and photosynthesis normalized by individual, which is very unusual and prohibits comparison with other studies.

Calcification rate and net oxygen production are quantitative very accurate parameters. As such, they were chosen for this study that aimed to recognize differences between treatments and between species. Normalizing these parameters by individual (as done by Evans et al., 2015, Titelboim et al., 2019) is meant to be more informative and more comparable than other ways of presenting this data that in many cases doesn't include the amount of foraminifera in the experiment or when normalized to mg is less specific because it can represent either more small specimens or less adult specimens that will have different growth rates.

It is also less precise if not all individual were of the same size (which they most likely were not).

Specimens were confined in size as mentioned in the manuscript in the method section. We have now added to the text the specific size fraction used for the experiments.

And I wonder, if you measured photosynthetic activity via oxygen production, why didn't you simply also measure respiration via oxygen consumption in darkness? This would have provided another valuable indicator for holobiont condition, and would have allowed to calculate gross oxygen production. So far, you only provide net values for photosynthesis, not considering that respiration is likely to be higher under high temperatures, which results in lower net photosynthesis, even though the actual production of oxygen may be constant.

Indeed. We have measured net photosynthesis. The use of this parameter clearly takes into account the fact that respiration lowers net photosynthesis. We understand from this comment that this was not clear and we adjusted the text to always say "net oxygen production/photosynthesis".

Lastly, I think this study could further be improved by discussing some interesting observational details such as bleaching (Did you observe it? In which species? Mottled or more gradual? Or mortality? Reproduction?) as well as by including time into the statistical evaluation of the results (e.g. two-factorial analysis of variance with time and temperature as factors).

There were no related observations of bleaching, mortality, reproduction etc (otherwise they would have been mentioned). This is in fact another indication that the parameters examined in this study are the most appropriate ones as their sensitivity can identify stress before a fatal response was reached. We added to the revised version additional statistical analyses, including two-factorial analysis of variance with time and temperature as factors.

Technical comments:

L2: "Roommate problems or successful collaboration?" this title sounds catchy, but the question is still as valid as before. . . I don't think the paper is resolving this question,

Following our response to comment 1 we believe that our title does describe the content of the paper and is not only valid but the formulation of it as a question actually emphasis the complexity of the relationship between the holobiont components.

 -- and also the first part of the title is very broad (e.g. climate change includes more than just temperature stress) and should be a bit more specific

We except this part of the comment and changed the first part of the title to: "Foraminiferal holobiont thermal tolerance under future warming...."

L11: ". . . hyaline diatom-bearing *Amphistegina lobi*fera and the proceallaneaous dinoflagellate-bearing *Sorites* . . ."

Changed

L12-13: see discussion above

Changed according to the comments of reviewer 1

L16-L17: "future warming will significantly shift the relative contribution. . ." this is taking the implications way too far. You only see small differences in their response to 35◦C in respect of timing. Please be more specific (in general) and stick to what you actually show.

This was changed to "may" according to the comment by reviewer 1. Further, the word "significantly" was deleted to make the sentence less affirmative

L18-21: You mention pre-exposure for the first time here, and it is rarely described in the manuscript in general. What do you mean by this? The season? Or different acclimation temperatures? I also don't understand why you suggest that it reduces thermal tolerance. Please reconsider these statements and adjust to your results and the discussion.

Following the review process, we have reconsidered this part of the manuscript and recognized that there are not enough evidences to make this case. Thus, we have decided to eliminate the results of the February experiment and the discussion regarding pre-exposure

L27-28: ". . . one of the regions most affected. . ."

Changed

L30: "Symbiont-bearing large benthic foraminifera . . ."

Changed

L34-35: To my knowledge, none of these studies really provided evidence for temperature control on symbionts composition. Some suggested that there may be a connection, but statements saying that they are "strongly controlled" would definitely require further proof, especially since other studies did show extremely flexible relationships (e.g. several Lee et al. studies, Pochon et al. 2007, Schmidt et al. 2016)

We have changed it to: "The symbiont composition of LBF was suggested to be controlled by temperatures"

L45: delete "calcifiers"

Deleted

L55-56: there are much earlier studies that describe these species in much more detail (such as Hansen & Burchard 1977 and Hottinger 1977) that deserve to be cited here

We added additional earlier references but we couldn't find the ones specifically mentioned

L59: ". . . Israel, during. . ."

Changed

L60: the picked size fraction is crucial when it comes to assessing growth/calcification as it is strongly linked to ontogenetic phase. Hence, please provide this information in the manuscript and not in the supplemental materials.

Added

L62: the same accounts for the sample sizes. It is important to know in order to judge the power of the study.

Added

And ". . . 60-ml airtight. . ."

Changed

L 68: there is still a lot of detail missing regarding the culture conditions: was there water flow? What kind of water were they in? Were they fed? pH? . . .

There was no water flow, this is implicit from the description of airtight Erlenmeyer that are important for measurement of oxygen and alkalinity. We use natural sea water as indicated in line 88. We do not feed them because we use natural seawater filtered above 45 µm, that contain the foraminifera and symbiont algae nutrition. pH of water through the experiment was added to the results.

L65: Which temperatures? In the baths or the flasks? What means regularly? Once a week or once per hour would both be regularly but are very different.

The flasks are incubated in the water baths and thus the temperature of the water within the flasks is controlled by the water in the bath. These were monitored using HOBO data loggers that recorded temperature every 1 hour. In the manuscript, the word "regularly" is now replaced with this description.

L71: Why did you use calcification rate as the only parameter? Please explain.

February experiment is now eliminated from this manuscript because of the differences in methodologies.

L74: Same for: why did you only include *A. lobifera* for the spring experiment?

February experiment is now eliminated from this manuscript

L77: Which "constant conditions"?

These details were at the supplementary file but are now written in the manuscript itself

L79-80: How many samples didn't show oxygen production? And any suggestion why?

A total of three samples didn't show normal oxygen production (this was detailed in supplementary and now moved to the main text) and in any case the number of replicates per treatment didn't decrease below three. The word " apparent " was replaced with " similar values of net oxygen production as other samples" which describe the scenario better, we cannot speculate why this happened.

L81: How was temperature adjusted? What instruments did you use to control this?

The temperature is controlled by heating circulators with a thermostat and were adjusted manually.

L82: where are these temperatures expected? In the Med Sea? The Red Sea?

Added "in the Eastern Mediterranean"

L85: Was the water filtered? Pre-conditioned temperature-wise?

Water were filtered through 45µm to ensure nutrition for the holobionts but also reduce noise in the oxygen measurement. Other than that, the water was not treated.

L85-90: Please give more details on the method. Did you do this in the 60-ml flasks? What was the time frame? Were foraminifera pooled?

Water from all the Erlenmeyer flasks were replaced at the same time, immediately transferred to air-tight syringes, and then measured for their alkalinity and dissolved oxygen. Measurements of oxygen were conducted immediately and of alkalinity within the next two days. To ensure no changes in water properties accurse in this time frame standard material is measured before and after the first and last sample of the set, respectively. Foraminifera were kept inside the Erlenmeyer flasks throughout the experiment.

L91-93: Same details are missing here, as well as references. What instruments did you use? What light? Which temperatures? ". . . μg L-1 ..." and please define RDO.

This follows the same protocol as described in the last comment. We added the technical details of the sensor, fixed "μg L-1" and defined RDO in the text.

What was the accuracy value?

The accuracy of the optical dissolved oxygen sensor was better than ± 0.01 mg/L

And in general, you normalize both parameters by specimen. For better comparability, they are usually normalized by species size (given by surface area or weight etc.). Please consider doing so.

See response to specific comment 4

L95-97: Please give n for each parameter, treatment, time point, described which data got transformed, and which tests exactly got chosen "accordingly".

We added all of this information to the revised manuscript in 2.3 Statistical analysis as a summarizing table

L99: " . . .cases where normality . . . non-parametric Kruskal-Wallis test . . ."

Changed

L100: Please name the "proper" post-hoc tests.

This is also added to the summarizing table in section 2.3 statistical analysis in the revised manuscript

L106: you cannot say whether there would be differences in *A. lobifera* between winter and spring, but sounds as if you do. Please rephrase.

February (winter) experiment is now eliminated from this manuscript

L108: What is the "x" in your box plots? Please indicate significant differences by a letter report. And give n in captions. Also provide full species names and specify what the whiskers represent (SE?).

Figure and caption changed as suggested

L113f: Please avoid expressions like "much higher", "a decrease" or "substantially different". What does that mean? Please provide statistical test results and/or how big is the difference (twice as high, ~20 %...)

This sentence was deleted as part of the exclusion of the winter experiment however this comment was noted for the rest of the revised manuscript

L120: "The symbionts' photosynthetic . . ."

The sentence is changed and how is the "sensitivity pattern" different? Apart from one week in the 35∘C treatment, they look very similar to me.

This is what we mean. We rephrased it to: "indicates faster response than that presented by calcification rates"

L122: As mentioned before, please provide overview of statistical results here, ideally in figure.

Added

L123: If you mention this "abnormal value" please state in which way it was abnormal and why you suspect this to be related to handling. You say it's not used for average and SE calculations. Does that also mean the further statistical analyses?

We added "abnormally high value" to indicate the way it is abnormal and considering the suggestion of reviewer 1 removed the statement about sample handling

Fig. 3: please jitter weeks more, so they are easier to recognize. In which week did you have the extreme value?

The graph is modified, and we also included the week of the extreme value in the legend and in the caption

L128: "significant negative response" I am not sure the word significant is used correctly here.

Removed the word "significant"

L134: "week" and what means "massive bleaching"? Please describe, give proportions etc.

This is added to the manuscript

L135-136: "... between 25∘C and 32∘C, and was thus clearly ..." As described in the general comments, please describe it more. What may be the reason for this bleaching? Were both species affected? *A. lobifera* could have a lower light tolerance than *Sorites* and could therefore bleach. Or since it only affected the lower temperature range, it could actually be related to reproduction. Why didn't you exclude the bleached specimens from the analyses? I suggest to include the data anyways, e.g. only for comparison in the supplementary, or to conduct analyses on those samples that had no bleaching. Moreover, as the 35∘C had no bleaching, please provide at least these values to compare with former weeks.

As suggested, this is added and discussed in the revised manuscript

L142-143: Well, I think "clear differences in thermal tolerance" is a bit exaggerated. I would call them rather subtle.

We deleted this as part of the first reviewer comments

L144: This is not true, the calcification seems lower at 35◦C (significances missing), at least in the winter populations. Plus, the experimental exposure in spring was shorter, so the response may have just been delayed, as suggested by the reduced oxygen production.

As mentioned before, we excluded the winter experiment since it was done in slightly different conditions than the spring experiment. Furthermore, the difference between the species is not just in the magnitude but also in the timing. *Sorites orbuculus* reduced calcification only after the second week, which indicate it is more resilient than *A. lobifera*. It is true that this is a short-term experiment and therefore (and following the comment of reviewer 1) had changed the sentence to be less affirmative.

L147: please rephrase, something here doesn't make sense. And I again don't agree that you can state that there is a "strong dependence", as many other factors have been shown to be at least as important. Please also include some newer references here.

Rephrased and added a newer reference

L149&L161: "dinoflagellates" and change to "Symbiodinium" to Symbiodiniaceae (as you seem to be aware this taxonomic system has been revised)

Changed

L153: ". . . control a holobionts ..."

Changed to "... control the holobionts..."

L154-155: What "mechanism to cope with thermal stress was observed"? Please describe, I do not think this paper actually showed 'shuffling'.

Rephrased, observed was changed to suggested. We deleted 'shffuling' and better explained the finding of Schmidt

L156: ". . . explain the observation . . ."

Changed

L159: ". . . describes ..."

Changed

L164: "... 35◦C, whereas in ..."

Changed

L165: ". . . inhibited, indicating that it is. . ."

The sentence was completely changed following comment by reviewer 1

L166: That means you refer to *Sorites* only? When is "earlier"?

Does this refer to old line 162? If so, yes this refer to the symbionts of *Sorites*. Earlier, is from the first measurement after one week contrary to calcification that only decrease after the second week. This is now clarified in the text.

L168: An apparent higher sensitivity (earlier/stronger response of the symbionts than the holobiont) was also observed by other studies such as Prazeres et al. 2017, Stuhr et al. 2017, Schmidt et al. 2016

References added

L170: Here you mention bleaching again: so were these specimens that you measured calcification on already bleached? If there was, this may also indicate that there was another stress factor such as too high light intensity or the wrong light spectrum (e.g. Hallock once showed that blue light facilitated high growth rates but at the same time led to bleaching). Please discuss your observations.

No, these were not bleached. However, the decrease in photosynthesis is considered as a negative response of the symbiont algae and thus compared to the observation made by Hallock regarding bleaching. We added clarification of this in the text.

Light intensity in the experiments is not a factor causing bleaching since for *Amphistegina* this light was used before in published (Titelboim et al., 2019) and unpublished experiments. For *S. orbiculus*, this is not the case since through the 3 week experiments no bleaching was observed.

L172: The resolution is a very important point! One week is a long time for a foraminifera! Plus, so far I don't even know how long your calcification measurements or photosynthesis measurements took. They don't calcify continuously all day long, so the time frame may strongly influence the results. The same accounts for the photosynthesis, which varies over the time of the day.

Indeed, the weekly resolution limits our ability to identify changes in a shorter-term time interval. However, the aim of this study is not to quantify changes in the daily cycle of calcification and photosynthesis but to examine the response over longer time intervals (measurement per week).

The description of handling time and measurement is not relevant to this point and hopefully this was clarified in the response to comment on lines 85-90, and also in the text in the method section.

L174: Please specify the time of the onset.

This part was deleted with the discussion on pre exposure

L175: What do you mean by "very cold"? I think that is very relative... give a tem- perature range of what is usually encountered in the Med Sea in winter and spring, and ideally state what were the temperature measured during sampling in the methods section.

This part was deleted with the discussion on pre exposure

L178: ". . . symbionts, or both. However, while the *A. lobifera* spring . . . "

L180: In which way did they respond "negatively"? Please be a bit more specific so the reader does not have to go back to each of the studies you cite to find you what you mean, and "... that, while ..."

This part was deleted with the discussion on pre exposure

L181: ". . . temperatures benefits the . . ."

This part was deleted with the discussion on pre exposure

L186-187: There have been studies modeling the future changes in distribution and hence contribution, which should be cited here (e.g. Weinmann et al. 2013, 2017)

References added

L187: If you mention pre-exposure here (as in the abstract) please elaborate a bit more in which way they had different pre-exposures in the methods as well as the discussion parts.

Pre exposure is no longer discussed in this manuscript

L188-189: I don't understand this statement at all. Why? Where is you evidence for that?

This part was deleted with the discussion on pre exposure

L189: Again, I find "clearly shows" a bit exaggerated.

"Clearly" is deleted

Supplement 1: Is the number of replicates given the value before or after exclusion of same samples? Why are the numbers different for the different time points? How did you deal with this unbalanced design in your statistical analyses?

This has been moved to the main text of the revised manuscript. Regarding the unbalanced design: an unequal sample size is only a problem if it validates the homogeneity of variance assumption. Since ANOVA is considered robust to some departures from this assumption it is common to use it even if the number of samples is not similar in all treatments. Specifically, this is not an issue with our data since this assumption is valid with high significance in all cases.

You used once filtered and once unfiltered water. Why? And why did you pre- condition the spring *Sorites* to another temperature than the rest? This is very crucial information and must not be excluded from the actual manuscript! I am not sure if you can compare your data the way you do with all these differences.

Considering these differences, February experiment and the comparison between spring and winter populations was excluded from the manuscript. Each species was pre-conditioned to its specific optimal temperature.

Sometimes you give two numbers after the comma, sometimes six or other. . . please be consistent (and usually its three).

The supplement file was replaced, and this comment was followed in the new file.

Why do you give four stars (they are actually called asterisk)? Usually, these are used to indicate the level of significance, from one (lower end of significance) to three (highly significant).

Since this presentation of results is not clear we changed it to include p-values between all treatments

Why is some text red? Please explain in captions.

Significant results are marked in red. This is added to the captions of the new supplement

Table S2.2 and others: What are "1, 2 and 3" in your column headers? I cannot under- stand your statistical results if I don't know what is which group.

The number of the group is indexed in the left part of the table marked with { }. We believe and hope that this clarifies it.

Table S5.2: "Stars indicate homogenous groups and thus significant differences be- tween them"? This makes no sense to me, because if they are homogenous, they are similar, so no difference. . .

As mentioned in previous comments, since this presentation of results was not clear we changed it to include p-values between treatment

[revised manuscript text omitted]
  S4 S3.1 and S3.2). Net photosynthesis show faster and clearer response than that presented by calcification rates. 
[revised manuscript text omitted]